# Detection Transformer with Multi-Scale Fusion Attention Mechanism for Aero-Engine Turbine Blade Cast Defect Detection Considering Comprehensive Features

**DOI:** 10.3390/s24051663

**Published:** 2024-03-04

**Authors:** Han-Bing Zhang, Chun-Yan Zhang, De-Jun Cheng, Kai-Li Zhou, Zhi-Ying Sun

**Affiliations:** School of Mechanical Engineering, Jiangsu University of Science and Technology, Zhenjiang 212003, Chinachengdejun0126@just.edu.cn (D.-J.C.); szy611@just.edu.cn (Z.-Y.S.)

**Keywords:** aero-engine turbine blade, multi-scale defect detection, attention-based channel-adaptive weighting, multi-scale feature fusion, R-focal loss

## Abstract

Casting defects in turbine blades can significantly reduce an aero-engine’s service life and cause secondary damage to the blades when exposed to harsh environments. Therefore, casting defect detection plays a crucial role in enhancing aircraft performance. Existing defect detection methods face challenges in effectively detecting multi-scale defects and handling imbalanced datasets, leading to unsatisfactory defect detection results. In this work, a novel blade defect detection method is proposed. This method is based on a detection transformer with a multi-scale fusion attention mechanism, considering comprehensive features. Firstly, a novel joint data augmentation (JDA) method is constructed to alleviate the imbalanced dataset issue by effectively increasing the number of sample data. Then, an attention-based channel-adaptive weighting (ACAW) feature enhancement module is established to fully apply complementary information among different feature channels, and further refine feature representations. Consequently, a multi-scale feature fusion (MFF) module is proposed to integrate high-dimensional semantic information and low-level representation features, enhancing multi-scale defect detection precision. Moreover, R-Focal loss is developed in an MFF attention-based DEtection TRansformer (DETR) to further solve the issue of imbalanced datasets and accelerate model convergence using the random hyper-parameters search strategy. An aero-engine turbine blade defect X-ray (ATBDX) image dataset is applied to validate the proposed method. The comparative results demonstrate that this proposed method can effectively integrate multi-scale image features and enhance multi-scale defect detection precision.

## 1. Introduction

Turbine blades serve as critical components of aero-engines, constantly operating under high temperature, high pressure, and corrosion conditions [1]. Under such circumstances, the various components of aero-engines are prone to degradation, including cracks, ablations, and pits, thereby affecting flight safety. Therefore, detecting defects in turbine blade casting before aero-engine assembly plays a crucial role in improving aircraft performance. This not only significantly extends the service life of aero-engines but also helps avoid secondary damage to the blades in harsh environments [2].

Due to the complex structure of turbine blades, it is inevitable that casting defects, such as slag inclusions, cracks, gas cavity, and cold shuts, will be developed during the casting process [3]. For the aforementioned defects, Non-Destructive Testing (NDT) is generally used for defect detection since it does not damage the physical structure of the turbine blade. This includes ultrasonic testing [4], eddy current testing [5], radiographic testing [6], infrared thermography [7], and magnetic particle testing [8].

For instance, Yang et al. [9] applied ultrasonic infrared thermography to detect the impact damage of a Carbon Fiber-Reinforced Plastic (CFRP) specimen for Unmanned Aerial Vehicles (UAVs), proposing defect-merging algorithms incorporating both time- and space-domain methods with a few thermal images. Moreover, Yin et al. [10] analyzed the trajectory of the Lissajous Curve (LC) of impedance data during eddy current testing presented on the complex plane, proposing a new LC analytical model to effectively extract features of the LC graph, enabling automatic defect identification. In addition, Jamison et al. [11] employed full-field infrared thermography imaging technology to monitor AlSi10Mg samples during SLM production, successfully applying it for statistical analysis of system processes during SLM production errors. In addition, Li et al. [12] proposed frequency-band-selecting pulsed eddy current testing, based on a frequency selection strategy. This technique improved the sensitivity of traditional pulsed eddy current testing for local wall thinning defect detection. Moreover, Balyts’ kyi et al. [13] proposed estimating the untightness of the cylinder of an internal combustion engine, dependent on the wear of the sealing ring of a piston, according to the value of its gap, volume, and rate of gas scavenging to the crankshaft box. Furthermore, Hossam et al. [14] applied laser-generated ultrasonic waves to realize the remote localization and detection of two-dimensional defects. The experiments proved that the proposed system combined the advantages of photons and ultrasonic waves, thus achieving two-dimensional visualization of defects.

To address the issue of fake magnetic particle indicator point noise in magnetic particle testing results, Chen et al. [15] introduced Hough transform to automatically identify the dominant features of magnetic particle indicators, effectively identifying scratches on hub bearings. Although the above traditional NDT methods have proven useful for defect detection, they still rely on specialized equipment and prior knowledge, greatly limiting the development of automation for defect detection technology.

Currently, object detection, as a branch of deep learning, is widely used in defect detection in industrial fields [16]. Existing object detection models based on Convolutional Neural Network (CNN) architecture can be divided into two categories: two-stage methods (Fast R-CNN [17] and Faster R-CNN [18]) and one-stage methods (YOLO [19,20,21,22,23,24,25] and SSD [26]). Two-stage methods are characterized by their small object detection tasks with high precision requirements. For instance, Hu et al. [27] applied ResNet50 to replace the original backbone network of Faster R-CNN for efficient feature extraction and incorporated the residual unit of ShuffleNetV2 to improve detection speed and accuracy. Moreover, Cheng et al. [28] proposed a DS-Cascade RCNN with deformable convolution to detect hub defects, allowing adaptive adjustments of the position and size of the convolution kernel considering the defect shape; however, this process will increase the computational complexity of the model and it is not conducive to training. Furthermore, Gong et al. [29] introduced a transfer learning object detection model based on domain-adaptive Faster R-CNN. The proposed system achieved accurate detection of small-size void and inclusion defects in X-ray images of spacecraft composite structures. Yet, due to the constraints of a multi-stage process and redundant computation, two-stage models have some distinct disadvantages regarding detection speed. In contrast, one-stage methods can effectively address the issue of the speed of detection. For example, Liu et al. [30] proposed a weather domain-integrated network aimed at the task of multi-domain insulator defect detection and classification. This technique extracts cross-modal discriminant information, thereby improving the accuracy of detecting small targets. Moreover, Liu et al. [31] enhanced YOLOv7, integrated a new convolution operator into the backbone network, and extended the sensitivity field; thus, they reduced the parameters by 18.03% and computed the load of the defect detection model that was equal to 20.53%. In addition, Zhu et al. [32] proposed a variety of SSD-based modules to detect concrete cracks, where the feature fusion-enhanced module can effectively fuse the feature channel information to enhance object detection precision. In addition, Zhao et al. [33] proposed an entire RDD-YOLO network based on the improvement of the backbone network and the design of a dual-feature pyramid network. This process enhanced the representation richness and ensured low-dimensional feature utilization. Moreover, Yang et al. [34] proposed the combination of an Electrode-Grounded Droplet-based Electricity Generator (EG-DEG) and a Graphene Sheet-Embedded Carbon (GSEC) electrode to monitor droplet velocities on different triboelectric surfaces. In addition, Li et al. [35] proposed a model consisting of the combination of an SSD meta-structure and the lightweight network MobileNet. As a result, it optimized the structure of the SSD without sacrificing detection precision. Concerning Xu et al. [36], they proposed an improved YOLOv5 model with a CA attention mechanism, SIOU loss function, and FReLU activation function. This new combination improved the ability to capture low-sensitivity spatial information. To sum up, these CNN-based models can efficiently extract image features and automatically learn feature representations, excelling in handling spatial information in images. However, all listed methods face challenges in capturing multi-scale object information due to their limited receptive fields and complex anchor box generation strategies. As a result, CNN-based models frequently exhibit unsatisfactory multi-scale object detection capability in real industrial scenarios. 

Compared to traditional CNN-based methods, the transformer-based method has apparent advantages in global perception, long-distance dependency modeling, and multi-scale detection; thus, it benefits from the introduction of innovative self-attention mechanisms. DETR [37], being the most representative work of transformer-based models, simplifies the object detection process and effectively removes some hand-designed components. As a result, this technique achieves real end-to-end object detection with the global modeling capability of transformers. For instance, Dang et al. [38] proposed an efficient and robust sewer defect localization system based on DETR, applying the transformer’s self-attention mechanism to analyze sewer defect level and scope. Although object detection-based models have achieved higher detection precision and efficiency while reducing manual experience dependence due to the limitations of industrial complex environments, three crucial problems still reside in defect detection:(1)Due to the limited quantity of casting defects and the extremely imbalanced distribution of sample categories, it becomes challenging to collect sufficient data for model training to mitigate the risk of overfitting;(2)The complexity and variability in the size of casting defects pose significant challenges for the multi-scale object detection capabilities of existing object detection models. However, most current methods struggle with extracting multi-scale features, leading to a deficiency in mining deep semantic information and accurately locating objects;(3)Traditional defect detection methods rely on hand-designed components to improve detection precision, often based on extensive prior knowledge. However, this approach imposes significant limitations on their application across diverse industrial scenarios.

To address the aforementioned challenges, a novel DETR technique with a multi-scale fusion attention mechanism for aero-engine turbine blade defect detection is proposed. This method considers comprehensive features, enabling multi-scale defect detection even with imbalanced datasets. A JDA method is introduced, combining Mix-Mos, Mixup, and Mosaic techniques to effectively enhance the quantity and diversity of the dataset. Additionally, an attention-based channel-adaptive weighting (ACAW) module is constructed to fully use the complementary information across feature channels, leading to the elimination of redundant feature information. Then, an MFF module is introduced to optimize the interaction between feature mappings. Finally, the R-Focal loss is developed in the MFF attention-based DETR, applying a random hyper-parameter search strategy to accelerate the model’s convergence. An ATBDX image dataset is used to validate the proposed method. The comparative results demonstrate that the proposed method can efficiently integrate multi-scale image features and improve multi-scale defect detection precision when considering imbalanced dataset conditions. Therefore, the novelties of this work are summarized as follows:A novel JDA method is proposed based on the combination of Mix-Mos, Mixup, and Mosaic techniques. This method effectively enhances the quantity and diversity of the dataset. Compared to traditional random data augmentation methods, the proposed method can solve imbalanced dataset problems and avoid the overfitting issue during model training. Therefore, this will improve model prediction, precision, and generalization;The ACAW feature enhancement module is introduced for mining the dependencies among different feature channels. This module ensures focusing on necessary features while removing redundant ones, leading to further refining feature representations;A novel MFF module is proposed to integrate high-dimensional semantic information with low-level representation features in response to various defect sizes. In contrast to simply concatenating high-dimensional and low-dimensional feature map methods, the proposed module achieves effective MFF due to the optimization of the interaction between feature mappings, significantly enhancing the precision of multi-scale defect detection;An R-Focal loss is developed in the MFF attention-based DETR by applying a random hyper-parameter search strategy to accelerate the convergence of the model. Moreover, the training weights of the positive and negative samples, in addition to the easy and hard classification samples, are adaptively adjusted to further address the challenge of imbalanced datasets;To validate the proposed method, an ATBDX dataset is used for defect detection. The strengths of the proposed framework are demonstrated by comparing it to other methods in terms of diverse evaluation indicators. The comparison results highlight the better detection performance of the proposed method with respect to other traditional methods. This proves that this method not only alleviates the defect detection missing issue caused by data imbalance but also effectively enhances the multi-scale defect detection precision under imbalanced dataset conditions.

To sum up, the remainder of this paper is structured as follows: Section 2 introduces the relevant basic theories of the DETR model and the attention mechanism. Section 3 details the proposed defect detection method. Section 4 provides the implementation and showcases the ablation experiments results. Finally, the conclusions are drawn in Section 5.

## 2. Preliminaries

### 2.1. Detection Transformer

DETR does not adopt the anchor-based architecture or proposal-based architecture, but directly apply the global modeling capability of the transformer to regard the object detection problem as a set prediction problem. To ensure that there is a one-to-one correspondence between predicted and ground truth boxes, DETR chooses to apply the hungarian algorithm to perform bipartite graph matching on predicted boxes set and ground truth boxes set, then calculate the loss between each matched box. Accordingly, the one-to-one loss calculation can avoid the confusion in matching predicted and ground truth boxes, which makes the training process of DETR more stable and controllable, thus realizing end-to-end detection.

The overall structure of DETR is illustrated in Figure 1, which mainly contains three key components: a backbone for compact feature representation extraction, a transformer’s encoder-decoder, and the feed-forward network (FFN) that finally outputs the category and position of the predicted boxes.

#### 2.1.1. Backbone

Due to the high computational complexity resulting from the self-attention mechanism of a transformer, it is not suitable for directly extracting image features, especially with large-scale images. Therefore, a CNN, with lower computational overhead and superior image feature extraction performance, serves as the backbone for extracting image compact features. These can enable the subsequent transformer encoder module in the DETR to focus on learning the spatial dependencies required for object detection tasks at the feature level. Generally, the initial input image and the lower-resolution activation map generated by the CNN backbone can be expressed as $x_{img}\in R^{3\times H_{0}\times W_{0}}$ and $f\in R^{C\times H\times W}$, respectively. In this work, typical values used in the DETR were C=2048, $H=\frac{H_{0}}{32},and\;W=\frac{W_{0}}{32}$.

#### 2.1.2. Transformer

The transformer encoder in the DETR shares similarities with the standard transformer, comprising a multi-head self-attention (MHA) module and a Feed-Forward Network (FFN). However, as the original transformer design targets sequence data processing, a 1×1 convolution is employed to reduce the channel dimension of the high-level activation map f from *C* to a smaller dimension *d*. The operation generates a three-dimension feature map $z_{0}\in R^{d\times H\times W}$ that transforms $z_{0}$ into a two-dimensional feature vector $z_{0}\in R^{d\times HW}$. Moreover, to incorporate positive information into the image features extracted by convolution, the DETR incorporates positional encoding into the input of each attention layer. This approach enhances the model’s ability to learn the spatial positioning of ground truth boxes within the image space.

The transformer decoder in the DETR follows the standard transformer architecture, employing both the MHA and encoder–decoder attention mechanisms. Given the absence of a fixed-order relationship between the semantic features of the image and object instances, the DETR’s transformer decoder decodes *N* objects in parallel. It conducts reasoning and prediction based on the relationships between these objects. In addition, the DETR incorporates different positional encodings for each input, calculated by MHA. These learned positional encodings, termed object queries, are updated according to the decoder’s computations to provide updated object location information. Ultimately, *N* object queries are decoded via FFN into the coordinates and categories of the prediction boxes. This approach enables the DETR to establish spatial dependency between any two object instances in the feature sequence and generate multiple prediction boxes for different object instances in parallel.

#### 2.1.3. FFN

The DETR’s final prediction network comprises a three-layer Multi-Layer Perceptron (MLP) and a linear projection layer. The FFN predicts the center coordinates, height, and width of each object, while the linear layer predicts the object category. Given that the value *N* set by the DETR is fixed and typically much larger than the number of objects in the images, many predicted boxes that do not align with the ground truth boxes are classified as background classes, indicating no object detected.

### 2.2. Attention Mechanism

The attention mechanism is similar to the human visual attention process, initially scanning the entire image to identify regions requiring focused attention. Consequently, integrating this mechanism into detection models improves their perceptual capabilities towards objects, owing to accurate focus on critical regions and effective feature fusion. Furthermore, the attention mechanism allocates more attention to requisite regions, gathering pertinent information related to the target object and eliminating redundancy. The utilization of attention mechanisms in current object detection models falls into two categories: Channel Attention Mechanism (CAM) [39] and Spatial Attention Mechanism (SAM) [40], detailed in the following sub-sections.

#### 2.2.1. CAM

CAM models dependencies between feature channels and reweights features from all channels, thereby enhancing feature expression effectiveness. The Squeeze-and-Excitation Network (SE-Net) [39], a typical CAM implementation, exemplifies this process as shown in Figure 2. Initially, SE-Net compresses the input feature map $X\in R^{C\times H\times W}$ via global averaging pooling, yielding the channel representation vector $z\in R^{1\times 1\times C}$ containing global information for each channel. Subsequently, z undergoes transformation through two fully connected layers, generating the channel’s attention weight vector $S\in R^{1\times 1\times C}$ by modeling inter-channel dependencies. The final attention-enhanced feature map $X^{\prime }\in R^{C\times H\times W}$ results from multiplying the weight S with the input feature map X. Finally, SE-Net can obtain the importance degree of each feature channel through a self-learning process and output an enhanced feature map, effectively improving the model’s prediction precision.

#### 2.2.2. SAM

The SAM models dependencies between spatial positions in the feature map, enabling selectively focus on important regions while suppressing irrelevant position features. The architecture of SAM modules, illustrated in Figure 3, involves several steps. Firstly, the input feature map $X\in R^{C\times H\times W}$ is processed using two different pooling methods (average pooling and maximum pooling) to yield $X_{1}\in R^{1\times H\times W}$. These results are concatenated along the channel dimension to obtain $X_{2}\in R^{2\times H\times W}$. In addition, the number of channels is adjusted using a 1×1 convolution to produce $X_{3}\in R^{1\times H\times W}$, facilitating feature weight calculation through the Sigmoid activation function. Finally, the enhanced feature map is obtained by multiplying the original input feature map with the feature weight. Through the SAM, the model effectively utilizes contextual information from the image, thereby improving its predictive ability.

## 3. Proposed Method

In this section, the overall structure of the proposed defect detection method is introduced, as illustrated in Figure 4, comprising four aspects. Firstly, the JDA method is proposed to enhance the quantity and diversity of the dataset. Secondly, an ACAW feature enhancement module is introduced to mine dependencies among different feature channels, thereby refining feature representations. Next, a MFF module is proposed to integrate high-dimensional semantic information and low-level representation features, addressing various defect sizes. Finally, the MFF attention-based DETR with R-Focal loss is developed for detecting turbine blade defects on the ATBDX dataset. 

### 3.1. Joint Data Augmentation

For defect detection tasks, random data augmentation is commonly used to expand training data and address imbalanced dataset issues. Techniques such as random scaling, cropping, flipping, adding noise, color jittering, and contrast adjustment are employed. However, these methods often generate approximately duplicate data, introducing redundancy into the model. Additionally, they may result in the loss of detailed information such as image edges and textures, ultimately failing to effectively improve the model’s generalization ability.

To address these problems, the Mosaic method is proposed, involving the random selecting of *K* images from the dataset as original spliced pictures. Random data augmentation techniques, such as zooming and color gamut transformation, are then applied to each of the *K* images. Finally, the local regions of the images are stitched together and the coordinates of the ground truth boxes are updated. This approach enables the learning of more comprehensive and accurate feature information, compensating for distribution bias in the dataset and effectively mitigating the impact of data imbalance on model training.

In addition, the Mixup method, proposed by Zhang et al. [41], has an advantage in alleviating the overfitting problem caused by imbalanced datasets. Initially, this method selects two training images, $p_{1}$ and $p_{2}$, and their corresponding ground truth boxes’ coordinates, $y_{1}$ and $y_{2}$ (it is important to mention that the selected images should also be randomly augmented). Then, a random interpolation factor λ, ranging between 0 and 1, is set to control the interpolation proportion. Finally, $p_{1}$ and $p_{2}$ as well as $y_{1}$ and $y_{2}$ are linearly interpolated to obtain a new mixed image  p and mixed coordinates y, expressed as follows:(1)$$p=\lambda p_{1}+\left(1-\lambda \right)p_{2}$$
(2)$$y=\lambda y_{1}+\left(1-\lambda \right)y_{2}$$

Mixup serves as an image-level data augmentation method, effectively learning the dependencies between images and improving the model’s generalization ability and object detection precision. However, although the above methods can alleviate the overfitting problem caused by imbalanced data, their performance is limited on aircraft engine blade datasets due to the number of defects that present an extremely imbalanced issue.

To solve the above issues, a novel JDA method is proposed based on the combination of Mixup, Mosaic, and Mix-Mos. In this technique, Mix-Mos first performs Mixup on randomly selected images in the training set, and then performs Mosaic on the improved dataset. The proposed method can simulate a larger batch size without updating the hardware configuration to reduce the number of epochs required during the DETR training, effectively alleviating the slow convergence of the model. The flowchart of the proposed JDA is illustrated in Figure 5. The proportions of dataset images randomly selected by the Mixup, Mosaic, and Mix-Mos are defined as a%, b%, and c% respectively, where a≥b≫c. These value settings were selected based on constraints that will be explained clearly in Section 4. Compared to traditional random data augmentation methods, the proposed method can identify various combinations and dependencies among a large amount of image datasets to generate more robust feature representation. This will effectively improve detection precision under imbalanced dataset conditions.

### 3.2. Attention-Based Channel Adaptive Weighting

Although the CNN, serving as the backbone, can effectively extract high-dimensional semantic features of targets, it cannot match the features’ importance in different dimensions and convolutional channels. This results in having extracted features characterized by little multi-scale information and poor quality. Therefore, a novel ACAW feature enhancement module is developed to mine more multi-scale feature information and dependencies among various feature channels from the extracted feature map backbone, realizing better focus on necessary features while removing redundant ones. The architecture of the proposed ACAW feature enhancement module is displayed in Figure 6. 

For instance, the feature maps can be extracted from four convolutional layers of the ResNet50 backbone. In more detail, conv2_x, conv3_x, conv4_x, and conv5_x are generated in response to $L_{1},L_{2},L_{3}$, and $L_{4}$, respectively, as shown in Figure 6. Then, transforming all four feature maps into the same dimension and concatenating them together can be expressed as follows:(3)$$X_{0}=C\left(f\left(L_{1}\right),\;f\left(L_{2}\right),f\left(L_{3}\right),f\left(L_{4}\right)\right)$$
where $L_{i}(C_{i},H_{i},W_{i})$ denotes the different feature maps, f(·) represents the dimension transformation (up-sampling the feature layers through interpolation and down-sampling the feature layers through a 1×1 convolution) and batch normalization, and $C\left(\cdot \right)$ indicates that the feature maps, after dimension transformation, are concatenated along the channel dimension. 

Referring to Equation (3), the stacking feature map $X_{0}(C,H,W)$ can be obtained. Consequently, two successive convolutions are performed on $X_{0}$ to extract the channel information, as described in Equations (4) and (5):(4)$$W_{1}=GAP\left(Y_{1}\right)=GAP\left(C_{1}\left(X_{0}\right)+X_{1}\right)$$
(5)$$W_{2}=GAP\left(Y_{2}\right)=GAP\left(C_{2}\left(Y_{1}\right)+X_{2}\right)$$
where $C_{i}$ represents the i-th convolution and $GAP\left(\cdot \right)$ denotes global average pooling. The weight vectors $W_{i}(C,1,1)$ can be obtained by computing the global average pooling on $Y_{i}$. To obtain fusion weights with multi-channel feature importance, a simple MLP is designed to linearly map $W_{i}$:(6)$$X=X_{0}\cdot W_{F}=X_{0}\cdot \delta {(fc}_{2}(\vartheta {(fc}_{1}(W_{1}+W_{2})))$$
where ${fc}_{i}$ represents the i-th fully connected layer, $\vartheta \left(\cdot \right)$ denotes the ReLU functions, and $\delta \left(\cdot \right)$ is the Sigmoid activation, used to fix the fusion weights $W_{F}$, and varying between 0 and 1. Based on the above analysis, the channel-adaptive weighted feature map X(C,H,W) can be determined by multiplying $W_{F}$ with the original feature map $X_{0}$. This can fully utilize complementary information across feature channels, and further refine feature representations.

### 3.3. Multi-Scale Feature Fusion

The DETR applies the transformer’s self-attention mechanism to model long-distance dependencies and identify global contextual information, exhibiting good performance in large object detection. However, as the CNN backbone can only obtain high-dimensional semantic features through deep convolution, it is difficult to efficiently learn low-dimensional fine-grained features and apply information related to small objects, as they significantly affect the DETR’s detection performance. Therefore, to detect multi-scale defects of turbine blades, an MFF module is designed to integrate high-dimensional semantic information and low-dimensional representation features, as depicted in Figure 7. Compared to directly concatenating high-dimensional and low-dimensional feature maps, the MFF technique can achieve effective multi-scale feature fusion by optimizing the interaction between feature mappings, significantly enhancing the precision of multi-scale defect detection.

Referring to Figure 7, the input feature map X(C,H,W) can be obtained using the ACAW module, as described in Section 3.2, consisting of the enhanced multi-scale features. In addition, to capture the maximum and average features of the X, max pooling and average pooling were performed on the channel dimension of *X* to obtain $X_{max}(1,H,W)$ and $X_{mean}(1,H,W)$, respectively. Then, $X_{max}$ and $X_{mean}$ were concatenated along the channel dimension to obtain $X_{pool}(2,H,W)$, consisting of a fusion feature map carrying both the maximum and average information. Moreover, the specific process of transformation can be expressed as follows:(7)$$X_{pool}=C(X_{max},X_{mean})=C\left(GMP\left(X\right),\;GAP\left(X\right)\right)$$
where C(·) indicates that the feature maps after pooling are concatenated along the channel dimension, whereas $GMP\left(\cdot \right)$ and $GAP\left(\cdot \right)$ denote the global max pooling and global average pooling respectively. 

Consequently, to obtain the multi-scale feature weight $W_{M}(1,H,W)$, a submodule is designed to determine the high-dimensional mapping of $X_{pool}(2,H,W)$. This can be expressed as follows:(8)$$W_{M}=\delta (C_{3}(C_{2}(C_{D}(X_{pool}))))$$
where $C_{D}(\cdot )$ denotes the first deconvolutional layer, and $C_{2}(\cdot )$ and $C_{3}(\cdot )$ represent the second and third convolutional layers, respectively; moreover, the batch normalization is performed on the feature maps obtained by $C_{3}(\cdot )$ operation. Notably, these three convolutional layers adopt a 7×7 kernel size, resulting in a larger receptive field and fusing spatial information over a wider range. In addition, the 7×7 convolutional kernel size is characterized by fusing multi-dimensional features, enabling the learned feature mapping $W_{M}$ to express a much richer multi-dimensional semantic information. As a result, the Sigmoid activation function compresses the values of $W_{M}$ within the range of 0 and 1, meaning that the feature channels with larger values of $W_{M}$ contain more important feature information in the original input $X_{pool}$. Finally, the feature map $X^{\prime }(C,H,W)$ after performing the MFF technique can be calculated as follows:(9)$$X^{\prime }=F_{scale}\left(X,W_{M}\right)$$
where $F_{scale}(\cdot )$ represents the tensor multiplication between the enhanced feature map $X^{\prime }$ and the multi-scale feature weight $W_{M}$. Furthermore, the feature map $X^{\prime }$, obtained after MFF, simultaneously fuses high-level semantic information and low-dimensional representation features, enabling the subsequent transformer’s encoder–decoder to better understand the image content and object structure. This process would achieve accurate recognition and classification of multi-scale defect coordinates.

### 3.4. MFF Attention-Based DETR Defect Detection

Referring to Section 3.2 and Section 3.3, the proposed ACAW and MFF modules can significantly improve feature expression ability, resulting in a better DETR performance in multi-scale defect detection. However, because of the imbalanced sample distribution in the dataset, both proposed modules face challenges in learning the complex mapping relationships and defining the interactions between features. Although the proposed JDA method can expand the dataset and alleviate the data imbalance, a backpropagation mechanism is still required to focus calculations on a small number of categories during DETR training, resulting in effectively preventing the model from overfitting. The focal loss method, proposed by Lin et al. [42], optimized the traditional cross-entropy loss function to solve the imbalanced dataset issue. This method adaptively adjusts the training weights of the different samples to modify the focus of the model training. However, for DETR models, just applying the focal loss could lead to large fluctuations in the calculated loss and generate difficulties during the train process. Therefore, an R-Focal loss in the MFF attention-based DETR is developed. It applies a random search strategy to determine the hyper-parameters, significantly improving the training efficiency of the model. The procedure of the R-Focal loss calculation is illustrated in Figure 8, as it focuses on two important aspects (that will be detailed in the following subsections): 

(A)Controlling the weights of positive and negative samples; (B)Controlling the weights of easy-to-classify and hard-to-classify samples.

#### 3.4.1. Controlling the Weights of Positive and Negative Samples

During model training, the transformer’s decoder generates *N* candidate boxes in an image. Among these boxes, the candidate boxes contain positive sample objects whereas the rest consist of negative samples. The common cross-entropy classification loss tasks can be expressed as follows:(10)$$CE\left(p,y\right)=\left\{\begin{array}{c} -\log\left(p\right),\;\;\;\;\;\;\;\;\;\;\;\;\;\;if\;y=1\\ -\log\left(1-p\right),\;\;\;\;\;\;\;\;otherwise \end{array}\right.$$
where y indicates the true sample class. Particularly, when y=1, the sample belongs to the positive class. Moreover, p represents the probability that the model predicts the sample as a positive sample; this can be expressed as follows:(11)$$p_{t}=\left\{\begin{array}{c} p,\;\;\;\;\;\;\;\;\;\;\;\;\;if\;y=1\\ 1-p,\;\;\;\;\;\;\;\;otherwise \end{array}\right.$$

By substituting Equation (11) into Equation (10), the common cross-entropy loss can be rewritten as follows:(12)$$CE\left(p,y\right)=CE\left(p_{t}\right)=-log(p_{t})\;\;$$

To reduce the impact of negative samples, a balance factor $\alpha_{t}$ is added prior to the common loss function. As a result, the contribution of positive and negative samples to the loss through adjusting the value of $\alpha_{t}$ is controlled as follows:(13)$$CE\left(p_{t}\right)=-\alpha_{t}\log\left(p_{t}\right)$$
$$where{\;\alpha }_{t}=\left\{\begin{array}{c} \alpha ,\;\;\;\;\;\;\;\;\;\;\;\;\;if\;y=1\\ 1-\alpha ,\;\;\;\;\;\;\;\;\;otherwise \end{array}\right.$$

Similar to $p_{t}$, when y=1, $\alpha_{t}=\alpha$. However, for other values of *y*, $\alpha_{t}=1-\alpha$, where $\alpha_{t}$ and α vary in the range of 0 to 1. According to the above analysis, the focal loss calculation of controlling positive and negative sample weights can be summarized as follows:(14)$$\;CE\left(p,y,\alpha \right)=\left\{\begin{array}{l} -\log\left(p\right)\times \alpha ,\;\;\;\;\;\;\;\;\;\;\;\;\;\;\;\;\;\;\;\;\;\;\;\;\;\;\;\;\;if\;y=1\\ -\log\left(1-p\right)\times \left(1-\alpha \right),\;\;\;\;\;\;\;\;\;\;\;\;if\;y=0 \end{array}\right.$$
where a higher value of α showcases a greater emphasis on positive samples, while a lower value allocates more weight to negative samples. In general, α is typically set below 0.5 to balance the values of positive and negative samples in the loss calculation; therefore, it avoids scenarios where the weight of negative samples becomes disproportionately large to affect positive sample recognition.

#### 3.4.2. Controlling the Weights of Easy-to-Classify and Hard-to-Classify Samples

For instance, it is assumed that there are two samples (*S_A_* and *S_B_*) when predicting the samples’ category. In more detail, *S_A_* has a $p_{t}=0.95$ probability of belonging to category T, while *S_B_* has a $p_{t}=0.4$. Compared to *S_B_*, *S_A_* is more likely to belong to category T as it is considered an easy-to-classify sample. According to this case, it can be concluded that a larger $p_{t}$ for a certain category yields the ease of classifying the sample into that category. Therefore, $1-p_{t}$ is capable of determining whether it is an easy-to-classify sample or a hard-to-classify sample, and it can be modelled as follows:(15)$$CE\left(p_{t}\right)=-{\left(1-p_{t}\right)}^{\gamma }log\left(p_{t}\right)$$
where ${\left(1-p_{t}\right)}^{\gamma }$ refers to the modulating factor, and γ is used to adjust its magnitude. When $p_{t}$ is closer to zero, the modulating factor tends to the unit, indicating a larger contribution to the total loss. More particularly, when γ=0, the focal loss is transformed to a traditional cross-entropy loss. Therefore, controlling the weights of the easy-to-classify and the hard-to-classify samples in R-Focal loss can be summarized as follows:(16)$$CE\left(p,y,\gamma \right)=\left\{\begin{array}{c} {-(1-p)}^{\gamma }\times \log\left(p\right),\;\;\;if\;y=1\\ -p^{\gamma }\times \log\left(1-p\right),\;\;\;\;\;\;if\;y=0 \end{array}\right.$$

Moreover, the range values of γ are typically greater than zero, and a higher γ value indicates that the model has to pay more attention to difficult-to-classify samples. If the γ value becomes too large, the model may neglect easy-to-classify samples; thus, the recall value of the detection results must be reduced. Most current studies [43,44,45] often set the hyper-parameter values when applying the focal loss technique; however, this cannot be adjusted for different datasets, resulting in not reaching the detection precision optimum. Therefore, this paper develops an R-Focal loss technique to optimize the model’s detection precision. It is summarized as follows:(17)$$RF\left(p,y,\alpha ,\gamma \right)=\left\{\begin{array}{c} -\log\left(p\right)*\alpha *{\left(1-p\right)}^{\gamma },\;\;\;\;\;\;\;\;if\;y=1\\ -\log\left(1-p\right)*\left(1-\alpha \right)*p^{\gamma },\;\;\;otherwise \end{array}\;\right.$$
where the selection of hyper-parameters α and γ will affect the model’s performance and convergence speed, directly influencing the effectiveness of the focal loss module on the specific tasks. In this work, we propose an effective random selection strategy to set the best hyper-parameters of α and γ, and the specific selection procedure can be expressed as follows: 

**Step 1:** Set the random search ranges of α and γ as follows: $S_{\alpha }=\left(\alpha_{1},\alpha_{2},\cdots ,\;\alpha_{m}\right)$ and $S_{\gamma }=\left(\beta_{1},\beta_{2},\cdots ,\;\beta_{n}\right)$, where m and n represent the sizes of the hyper-parameter search sets;

**Step 2:** In the first T epochs of the model training, multiple groups of α and γ are randomly selected to train the model and calculate the R-Focal loss using Equation (17). In addition, the calculated loss value through each hyper-parameter group is recorded to evaluate the performance and convergence of the model;

**Step 3:** Based on the performance evaluation results of each 5 to 10 epochs, the worst-performing hyper-parameter combinations are eliminated, and the high-quality hyper-parameter combinations are retained;

**Step 4:** Repeat **Steps 2** and **3** to conduct multiple iterative training and hyper-parameter tuning to gradually approach the optimal parameter configuration;

**Step 5:** The hyper-parameters will no longer be adjusted until m≤1 and n≤1. Consequently, all subsequent DETR training will be performed according to the obtained optimal hyper-parameter configuration.

## 4. Experiments

### 4.1. Datasets and Metrics

In this study, the strengths of the proposed MMF attention-based DETR method are showcased by comparing it to other methods and experiments in terms of diverse evaluation metrics. Through image radiography, the obtained ATBDX dataset contains a thousand blade defect images. The full description of the ATBDX dataset and evaluation metrics are as follows.

**ATBDX dataset:** The original films of turbine blades were collected using conventional film radiography, guaranteeing that the ATBDX image dataset consists of real turbine blade images gathered from aero-engines. Then, to find defect blade images, original films were scanned through an industrial film digital scanner with optimal resolution scanning parameters to obtain multiple 2600 × 1200 pixel blade images as illustrated in Figure 9a. To further improve visible defects and provide sufficient pixel-level details, the blade defects were cropped to obtain 512 × 512 pixel images, as shown in Figure 9b. During the cropping process, the same defect was cropped nine times. The defect was placed in different positions of the image during each iteration, including the central position and the surrounding eight positions. As a result, the ATBDX dataset contains a total of 9682 images with resolutions no larger than 512 × 512 pixels, where a few defects near the image edges may have a width or height smaller than 512. Among these images, the number of each type of defect accounts for the proportion of the total dataset as listed in Table 1. Moreover, Figure 10 illustrates samples of various defect images. According to the characteristics of the ATBDX dataset, defects smaller than 10 pixels in size are categorized as small-size defects, accounting for approximately 30% of the total number of defects found in the dataset. Referring to the achieved analysis, it can be found that the ATBDX dataset is an imbalanced dataset, especially as the number of gas cavities and crack defect samples is too small for effective defect detection. Additionally, to ensure experiments’ fairness and randomness, the ATBDX dataset was divided into a training dataset and a test dataset using an 8:2 ratio. 

**Evaluation metric:** To verify the effectiveness of the proposed method, Precision, Recall, and *F*1-score were used to evaluate the performance of defect image classification. The equations modelling these three metrics are expressed as follows:(18)$$Precision=\frac{TP}{TP+FP}$$
(19)$$Recall=\frac{TP}{TP+FN}$$
(20)$$F1=\frac{2\times Precision\times Recall}{Precision+Recall}$$
where *TP*, *TN*, *FP*, and *FN* highlight the output results for True Positive, True Negative, False Positive, and False Negative, respectively. 

In this work, the proposed method utilizes ${AP}_{50}$ and ${AP}_{75}$ to evaluate the performance of defect localization. The mean average precision (mAP) is computed using the method adopted in Pascal VOC evaluation [46]; therefore, the detection bounding box is considered as *TP* if it has Intersection over Union (IoU)>0.5 (orIoU>0.75) with one ground truth box. To further evaluate the model’s multi-scale defect detection capability, ${AP}_{S}$, ${AP}_{M}$, and ${AP}_{L}$ were employed to test the average precision of detecting small-size defects, medium-size defects, and large-size defects, respectively. This can better showcase the enhancement in the multi-scale defect detection ability using the proposed method with the application of the ACAW and MFF modules.

### 4.2. Implementation Details

The hardware and software parameters of the experiment are listed in Table 2. During model training, Adam optimizers were utilized with learning rates of $1\times 10^{-4}$ and $1\times 10^{-5}$ to train the backbone and transformer, respectively, having a batch size of eight. A weight decay of $1\times 10^{-4}$ and a momentum of 0.5 were selected while the learning rate dropped every 50 epochs. In this test, ResNet50 [47] was chosen as the backbone to extract compact image features. Moreover, the transformer settings were as follows: the number of layers $N_{L}$ was set to six, the dimension of hidden layers in PWFFN $d_{ff}=2048$, $d_{model}=64$, the dimensions of object queries (the number of detection bounding boxes generated for each image) N=100, and the MHA head number h was equal to eight.

### 4.3. Ablation Studies

In this section, ablation experiments are conducted to showcase the contribution of each module to the proposed defect detection method. This study successively reports the impact of JDA, ACAW, MFF, and R-Focal loss modules on the multi-scale defect detection precision.

#### 4.3.1. Ablation Study for JDA

Compared to traditional random data augmentation methods, the proposed JDA method can identify various combinations and dependencies from a large number of image data to generate more robust feature representation. In this test, JDA was applied on the DETR model with the ResNet50 [47] module serving as the backbone, also known as DETR-R50. As described in Section 3.1, the proportions of Mixup, Mosaic, and Mix-Mos for random dataset images selection are defined as a%, b%, and c%, respectively. Therefore, multiple tests were performed to obtain the optimal values of a, b, and c. The specific experimental results are listed in Table 3 and Table 4. As illustrated in Table 3, c=0 was set at the initial experimental stage, meaning that the Mix-Mos method was not applied for data augmentation. Under this hypothesis, the values of AP, ${AP}_{50}$, and ${AP}_{75}$ increase as a increases, and the highest detection precision is achieved when reaching $\left(a,b\right)=(30,30)$. Compared to the initial baseline (only using random data augmentation) where ${AP}_{50}=86.3$, the obtained ${AP}_{50}$ due to the optimal selection of Mixup and Mosaic image ratios was improved by 1.5 points, demonstrating the effectiveness of the Mixup and Mosaic data augmentation methods in enhancing the model’s detection precision.

In the second experiments, to further improve the model’s detection precision, a relatively small range around the optimal value of ${AP}_{50}$ was set as $\left(a,b\right)=[\left(30,20\right),\;\left(30,30\right),(40,20)]$ and the value of *c* was computed for subsequent testing. Referring to Table 4, the model showed the best performance when $\left(a,b,c\right)=(30,30,5)$. Notably, ${AP}_{50}$ shows a downward trend while *c* increases, verifying that a larger c will lead to the model overfitting which may hinder the improvement of model defect precision. Through the second experiment, the proportions of a%, b%, and c% in the proposed JDA method were determined as 30%, 30% and 5%, respectively. More specifically, by applying the JDA method, the size of the ATBDX dataset was expanded from 9682 to 15,976. This allows the expanded dataset to learn more comprehensive and accurate feature information, helping to compensate the distribution bias present in the original smaller dataset.

Based on the above analysis, the optimal value of ${AP}_{75}=71.2$ was reached using the proposed JDA method, presenting 3.0 points improvement compared to the original model (${AP}_{75}=68.2$) that only utilizes random data augmentation. This proves that the proposed JDA method can effectively enhance the model’s capability to accurately classify and locate high confidence objects. Additionally, the values of AP and ${AP}_{50}$ were enhanced by 2.2 points and 2.3 points, respectively, indicating that the proposed JDA method makes a significant contribution to improving the model’s overall detection performance.

#### 4.3.2. Ablation Study for ACAW Module

The proposed ACAW feature enhancement module can mine for more multi-scale feature information and dependencies among various feature channels from the extracted feature map from the backbone, realizing the focus on necessary features while removing redundant ones. As described in Section 3.2, a 1×1 convolution is utilized to convert the channel numbers of feature maps $L_{1}$, $L_{2}$, $L_{3}$, and $L_{4}$ into 512. Consequently, to ensure a consistent feature map size was obtained, down-sampling $L_{1}$ and $L_{2}$ and up-sampling $L_{3}$ and $L_{4}$ using 1×1 convolutions and bilinear interpolation, respectively, were performed. This preserves both low-level image features and high-level semantic features. Then, to reduce the model’s complexity, channel dimension reduction was performed on the feature map $X_{0}$, converting the initial number of channels $C_{1}+C_{2}+C_{3}+C_{4}=2048$ into 512. At this level, a multi-layer attention mechanism was conducted on the input feature map $X_{0}$, and finally a 1×1 convolution was applied to convert the number of channels back to 2048 to confirm compatibility with subsequent network training. During model training, fine-tuning was employed to freeze the weights of the pre-trained model; thus, the added ACAW module was only trained within a few epochs, effectively accelerating the network training process while preserving the knowledge gained through the pre-trained model.

To investigate the ACAW module on the feature extraction capabilities of the different backbones, comparative experiments were conducted using three different backbones: (a) ResNet50 [47], (b) ResNet101 [47], and (c) VGG-16 [48]. For the first two, conv2, $conv3_{v}$, conv4_v, and conv5_v were selected in response to input feature maps $L_{1}$, $L_{2}$, $L_{3}$, and $L_{4}$ respectively. As for the third one, its last four convolutional layers served as inputs for ACAW. To reflect the complexity of the backbone, the parameters (*Params*) and GFLOPs of the backbone were calculated using an input size of 256×256, and the evaluation results are displayed in Table 5.

From Table 5, it can be observed that the mAP values of the three backbones shows a significant improvement through adding the ACAW module, demonstrating that this module can effectively improve the multi-scale feature extraction quality, and therefore enhancing the model’s defect detection precision. In more detail, the ${AP}_{50}$ of the DETR model using Resnet-101 reaches 89.3, presenting an improvement by 0.7 points compared to the DETR without ACAW. More specifically, the contribution of the ACAW module on the VGG-16 network is the most prominent as it presents an improvement of 3.3 points. Based on the above analysis results, it can be concluded that the DETR model with the VGG-16 backbone brings a significant enhancement in the object classification and localization performance after introducing the ACAW module, successfully solving the weaker feature extraction capability of the VGG-16 network issue.

#### 4.3.3. Ablation Study for MFF Module

In contrast to simply concatenating high-dimensional and low-dimensional feature maps, the proposed MFF module can achieve effective MFF through optimizing the interaction between feature mappings. This leads to significant improvement in the multi-scale defect detection precision. To validate the effectiveness of the proposed MFF module, defect detection performance is tested from two aspects: (a) multi-object defect detection and (b) multi-scale defect detection.

Therefore, Figure 11 plots the sample visualization of the multi-object defect results measured for three types of defects. The Box id represents the index of the matched box among the *N* bounding boxes. In addition, the second and third rows visualize the attention maps defined from the last encoder layer of the DETR and DETR-MFF (the input of the MFF module is changed from the output feature map of ACAW to the direct output feature map of backbone) models. This clearly identifies the models paying more attention to the region during the defect detection process. Moreover, referring to Figure 11, the DETR-MFF is able to capture the key shape and color information of defects better than the DETR, which can be evidenced based on the detection of redundancy defects and pseudo defects. When detecting the redundancy defects, the visualization of the DETR-MFF highlights more yellow pixels, highlighting that the model pays more attention to this region. This phenomenon is more obvious in the detection of pseudo defects. As shown in the pseudo defect detection results, the DETR only outlines the general contour of the defect, whereas the attention is shifted due to the impact of the defect slag inclusion as it is marked in the green boxed region. In contrast, the DETR-MFF module pays more attention to the turning points of the defect shape and depicts the specific shape through these points, enabling a better classification and localization of defects with complex shapes. In addition, the attention mechanism of the DETR-MFF cannot be shifted by multiple objects in the multi-object detection task, demonstrating the excellent defect localization capability and multi-defect detection performance of the MFF module.

To test the impact of the MFF module on the DETR’s multi-scale defect detection performance, small-size, medium-size, and large-size defects were selected for testing. Referring to the first and third columns in Figure 12, the DETR-MFF locates small-size defects more precisely compared to the DETR, which is proven by more yellow pixels concentrating on the defect region in the visualized attention maps. Moreover, from the second column in Figure 12, it can be seen that the DETR fails to connect the overall shape of large-size defects. In contrast, the DETR-MFF can capture the shape and brightness features of large-size defects in a better way, as illustrated in the red box at the bottom of the second column, thus predicting more precisely the coordinates of the bounding box. More specifically, when detecting medium-size defects, the bounding box produced by the DETR (green box in the fourth column) fails to locate the defect accurately due to the interference from other defects. This will lead to missed detections. In contrast, the DETR-MFF can optimize the interaction between feature maps through the attention mechanism, yielding to depict defect shape clearly.

To validate the effectiveness of the proposed DETR-MFF on the multi-scale defect detection capability, it was compared to the SSD [26], YOLOv5, and deformable DETR [44] methods. Meanwhile, the performance of each model in detecting small-size, medium-size, and large-size defects was evaluated using the pycocotools library [37]. Moreover, Table 6 lists the comparison results of mAP between traditional methods and the proposed DETR-MFF. It is clear that the DETR-MFF shows a significant improvement in all evaluation metrics compared to the traditional DETR method. In addition, the deformable DETR presents comparable performance with DETR-MFF on AP, ${AP}_{50}$, and ${AP}_{75}$ evaluation metrics, and demonstrates better detection capability for small-size defects. However, the DETR-MFF shows superior performance in medium-size defect and large-size defect detection, showcasing its stronger capability of multi-scale defect detection. Mainly for large-size detect detection, the DETR-MFF outperformed YOLOv5-L by 9.1 points on the ${AP}_{L}$ metric. This further highlights the DETR-MFF’s outstanding advantage in detecting large-size defects.

#### 4.3.4. Ablation Study for R-Focal Loss

The proposed R-Focal loss is able to adaptively adjust the training weights of positive and negative samples, as well as the easy-to-classify and hard-to-classify samples, to address dataset imbalance problems. Moreover, a random hyper-parameter search strategy is utilized to accelerate the model convergence. As described in Section 3.4, *m* = *n* = 5 was set by considering search time and cost. Furthermore, $S_{\alpha }=\left(0.1,0.25,0.5,0.75,\;0.9\right)$ and $S_{\gamma }=\left(0.2,0.5,1.0,2.0,5.0\right)$ were set according to the selection method defined in [42]. Then, a random search of the parameters was conducted within the first 30 epochs, and the parameter elimination was performed every 5 or 10 epochs. Finally, several values of R-Focal loss were obtained with the combination of different hyper-parameters as listed in Table 7. It can be observed that the values of R-Focal loss with $\left(\alpha ,\gamma \right)=\{\left(0.25,2.0\right),(0.25,5.0)\}$ are optimal below six at 30 epochs. In addition, from the comparison results of $\left(\alpha ,\gamma \right)=\{\left(0.25,1.0\right),(0.5,1.0)\}$, it can be concluded that α=0.25 represents the most significant influence on the loss. Therefore, based on the above analysis, the optimal R-Focal loss hyper-parameter is determined as $\left(\alpha ,\gamma \right)=(0.25,5.0)$. 

To demonstrate the advantage of the R-Focal loss in handling the imbalanced datasets, the DETR was trained using the conventional cross-entropy loss and optimal R-Focal loss. Calculations of the loss curve and recall curve with and without the R-Focal loss are displayed in Figure 13. Based on the highlight green box area, it can be observed that the DETR with R-Focal loss converges faster, due to the optimal selection of its hyper-parameter. Compared to the DETR when considering the cross-entropy loss, the Recall score of the DETR with R-Focal loss is improved by 1.93 points, indicating that the proposed R-Focal loss can improve the model’s generalization and enhance the learning ability for positive samples, thus contributing to handle the imbalanced dataset. Notably, the defect detection model with high Recall scores can significantly reduce the misdetection rate for blade defect detection, which is of great importance for aerospace industry applications.

Based on the above ablation experiments, the contribution of each module to the improvement of model detection performance was generated and was mainly reflected in ${AP}_{50}$ and Recall values. The improvement of AP and Recall varied between different modules, as shown in Figure 14. It can be observed that the MFF has the greatest contribution to ${AP}_{50}$, improved by 3.0 points. The reason is that the MFF module can enhance the model’s localization and classification capabilities on various defect sizes using an efficient MFF mechanism, thus significantly improving the precision of the multi-scale defect detection. However, since the MFF module introduces some False Positive samples while enhancing precision, the improvement of Recall is not remarkable. Therefore, the R-Focal loss is developed to solve the problem of imbalanced datasets in order to strengthen learning positive samples by adjusting sample weights during training. This does not only achieve higher Recall scores but also improves model generalization. Referring to the comparison results of Figure 14, the Recall score is improved by 1.9 points due to the constraints imposed by R-Focal loss, as it has the most remarkable improvement compared to other modules.

#### 4.3.5. Comparison with the-State-of-the-Art Methods

To evaluate the model’s capability in detecting multi-scale defects and handling imbalanced datasets, the detection of mAP for small-size, medium-size, and large-size defects was calculated using the two rarest defect categories in the dataset. Therefore, Table 8 lists the comparison results of detection mAP between the proposed method and other traditional methods. As a result, the proposed method has the greatest advantage in detecting large-size defects, achieving an ${AP}_{L}$ of 97.5, being 32.0 points higher than the worst method Faster-RCNN (${AP}_{L}=62.5$) and 6.8 points higher than the second-best method CenterNet (${AP}_{L}=90.7$). Although the original DETR has some advantages when considering large-size defect detection, the ${AP}_{L}$ is improved by 9.2 points through the proposed method, fully demonstrating the efficiency of the ACAW and MFF modules. For small-size defect detection, the proposed method outperforms YOLOv5-L by 5.5 points and the Deformable DETR by 4.8 points. While CenterNet has relatively small gaps with the proposed method in small-size and large-size defect detection, it still falls behind by 10.5 points in ${AP}_{M}$. Thus, based on the above analysis results, the proposed method is able to perform outstandingly in handling defects of various scales, and is capable of locating, in a more precise way, large-size defects.

As shown in Table 8, it is clearly found that the proposed method achieves superior performance in detecting the average precision of detecting gas cavities ${AP}_{GC}$ and crack defects ${AP}_{C}$, obtaining 92.7 and 88.6 for ${AP}_{GC}$ and ${AP}_{C}$, respectively. Moreover, YOLOv5-L has relatively poor detection precision on both defects, only achieving 40.3 and 41.8 for ${AP}_{GC}$ and ${AP}_{C}$, respectively. In addition, SSD, Faster-RCNN, and deformable DETR exhibit an obvious issue of detecting one defect category poorly when considering two defects. However, CenterNet has more stable detection performance but its ${AP}_{GC}$ and ${AP}_{C}$ are still lower, measuring 13.8 points and 8.2 points less than the proposed method. To sum up, the proposed method can effectively improve the model’s detection capability on imbalanced datasets using data augmentation and loss computation strategies.

To further demonstrate the superiority of the proposed MFF attention-based DETR model, YOLOv5, SSD [26], deformable DETR [44], Faster-RCNN [18], and CenterNet [49] were tested on the ATBDX dataset. Table 9 illustrates the defect detection performance of the different methods. It is clear that the proposed method can achieve the best performance on *F*1, Recall, AP, and ${AP}_{50}$ compared to other methods. The ${AP}_{50}$ value of the original DETR is 6.5 points lower than the proposed method, proving the effectiveness of the modules developed in this work. Compared to other methods, the ${AP}_{50}$ value of the proposed method is higher by 4.7 points and 2.2 points than the deformable DETR and the second-best method CenterNet, respectively. In terms of AP, the proposed method reaches 67.2, which outperforms SSD with VGG-16 backbone by 13.0 points and YOLOv5-L by 6.3 points. In addition, the high scores on AP and ${AP}_{50}$ demonstrate that the proposed method greatly performs at different IoU thresholds and has stable average precision for different defect categories, proving that the proposed method has good capability in detecting multi-scale and complex defects.

Moreover, it can be found that the proposed method achieves 80.8 in Recall score, which is largely attributed to the R-Focal loss and JDA. The Recall score of the proposed method is higher by 7.7 points and 1.6 points than the original DETR and SSD, respectively. The Faster-RCNN with ResNet-50 backbone is only 0.3 points behind the proposed method on Recall score, but lags far behind in other evaluation metrics, especially as it is 11.8 points lower than AP. Although CenterNet achieves the closest results to the proposed method, it still falls behind by 4.4 points in *F*1 due to its inferior capability in differentiating between two easily confused defects, e.g., pseudo defects and slag inclusion. In summary, the proposed method demonstrates higher defect detection precision and model robustness, which can adapt to various defect categories and data distributions.

Finally, the proposed method primarily targets pre-service inspection and maintenance of aero-engines, involving defect detection on engine blades after the casting pouring process is completed. By applying this technique to real-world aero-engine manufacturing, we can effectively reduce the frequency and time spent on engine repairs. The average service lifetime of each aero-engine is also extended by about 30%, promoting greater automation in defect detection across the aviation industry.

## 5. Conclusions

The existing methods have limitations in effectively detecting multi-scale objects and handling imbalanced datasets, which are challenging to apply in blade defect detection of industrial scenarios. Therefore, this paper proposes a novel multi-scale fusion attention mechanism-based defect detection method to address these limitations, integrating the JDA module, ACAW feature enhancement module, MFF module, and MFF attention-based DETR. A JDA method is built to enhance the quantity and diversity of the dataset, which not only solves the imbalanced dataset issue but also avoids the overfitting phenomenon during the model training. An ACAW feature enhancement module is introduced for mining the dependencies among different feature channels, realizing the focusing on necessary features while removing redundant ones, thus further refining feature representations. Consequently, an MFF module is proposed to integrate high-dimensional semantic information and low-level representation features, achieving multi-scale feature enhancement fusion to enhance the precision of multi-scale defect detection. 

Concerning the multi-scale object detection efficiency, the R-Focal loss is developed in the MFF attention-based DETR to accelerate the convergence of the model. Meanwhile, the training weights of the positive and negative samples, as well as the easy and hard classification samples, are adaptively adjusted to further solve the imbalanced datasets issue. The advantage and effectiveness of the proposed method are demonstrated by comparing it to other methods in terms of diverse evaluation indicators. The comparison results show that the proposed method has better detection performance than other traditional methods. This proves that the proposed technique has the ability to consider the diversity of imbalanced datasets to develop a reliable detect defection model, being of great significance for turbine blade casting quality decision-making in the aerospace industry.

## Figures and Tables

**Figure 1 sensors-24-01663-f001:**
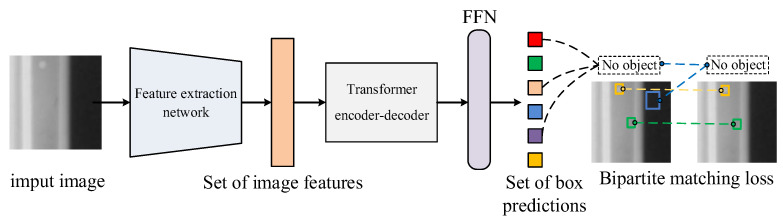
Overall structure of DETR.

**Figure 2 sensors-24-01663-f002:**
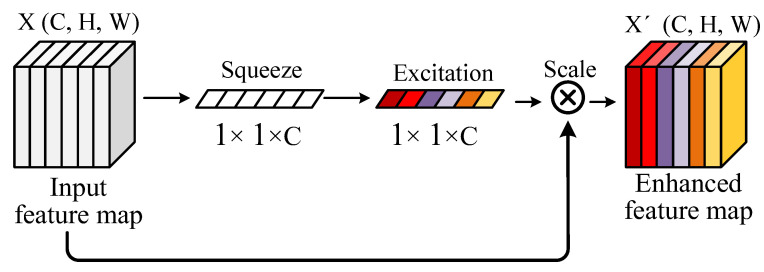
The architecture of the SE-Net.

**Figure 3 sensors-24-01663-f003:**
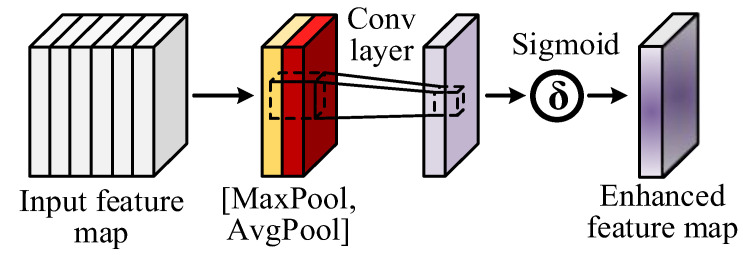
The architecture of the SAM.

**Figure 4 sensors-24-01663-f004:**
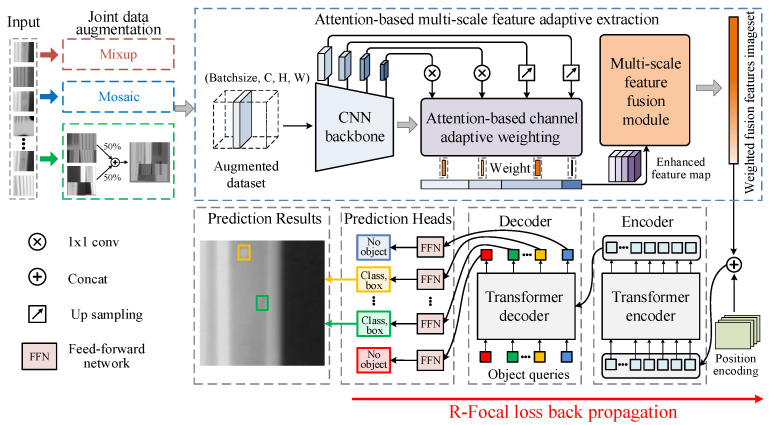
The overall architecture of the proposed MFF attention-based DETR.

**Figure 5 sensors-24-01663-f005:**
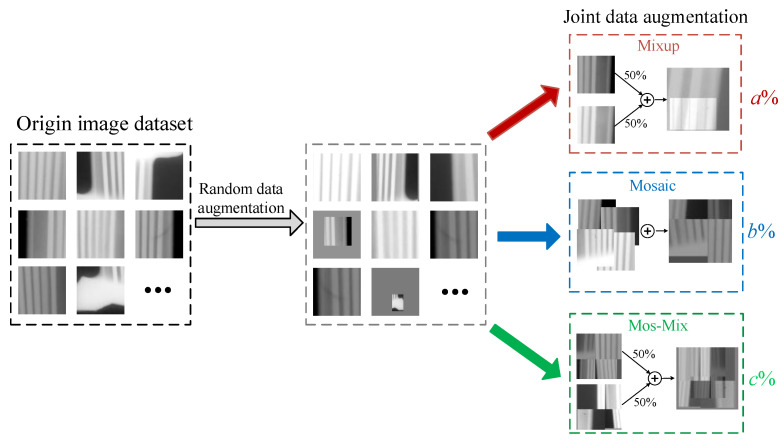
The flowchart of the JDA.

**Figure 6 sensors-24-01663-f006:**
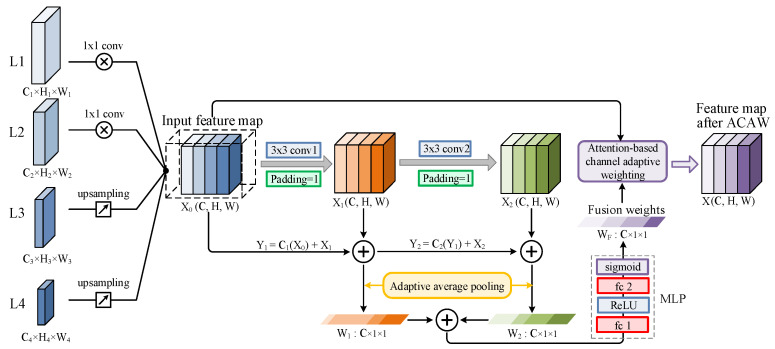
The architecture of the ACAW feature enhancement module.

**Figure 7 sensors-24-01663-f007:**
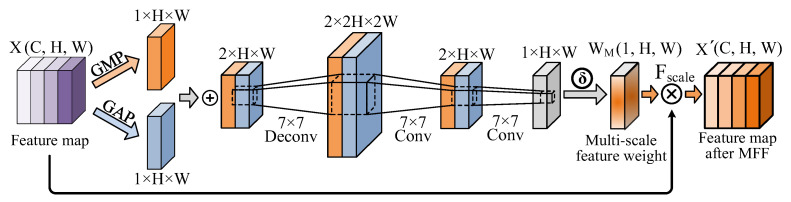
The architecture of the MFF module.

**Figure 8 sensors-24-01663-f008:**
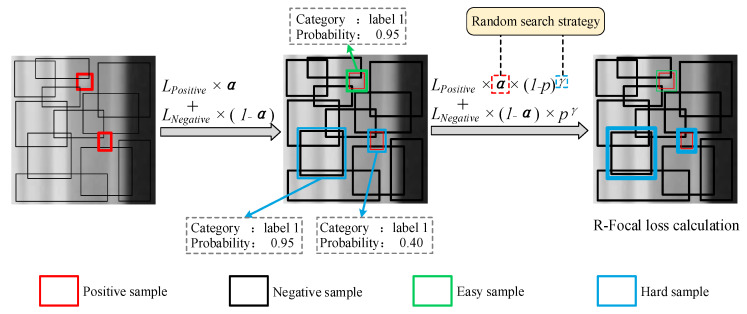
The procedure of R-Focal loss calculation.

**Figure 9 sensors-24-01663-f009:**
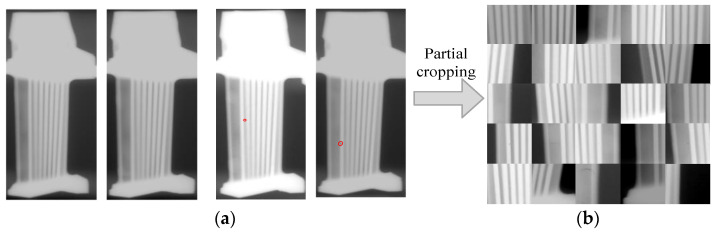
X-ray images of aero-engine turbine blades. (**a**) Original images (**left**: non-defect turbine blades; **right**: defect turbine blades); (**b**) defect turbine blade after cropping.

**Figure 10 sensors-24-01663-f010:**
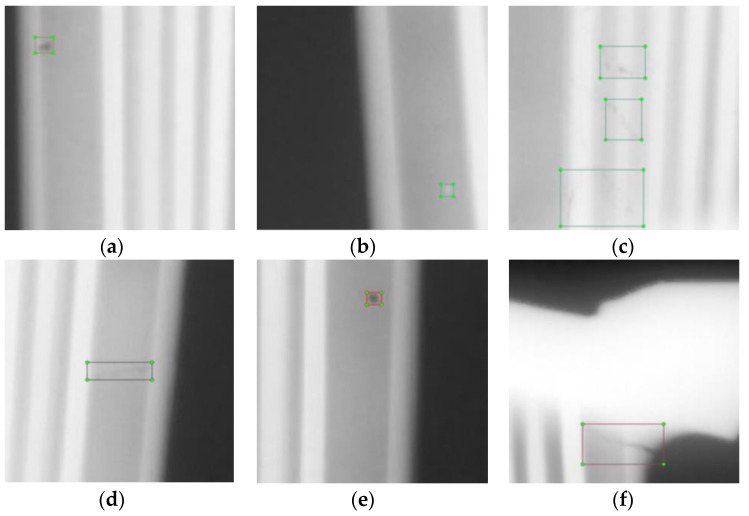
Each type of blade defect in the dataset. (**a**) Slag inclusion; (**b**) redundancy; (**c**) pseudo defect; (**d**) cold shut; (**e**) gas cavity; (**f**) crack.

**Figure 11 sensors-24-01663-f011:**
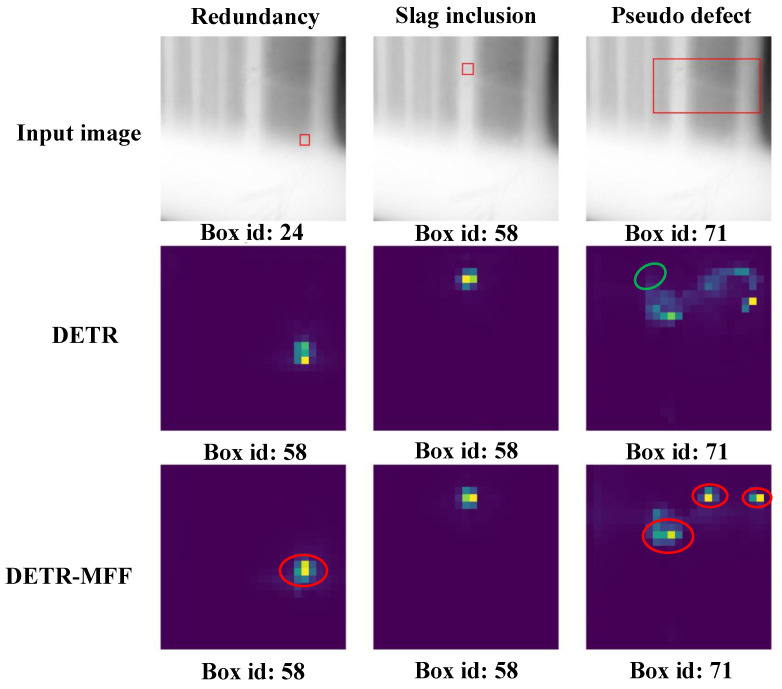
Sample visualization of the multi-object defect analysis from the DETR and DETR-MFF models.

**Figure 12 sensors-24-01663-f012:**
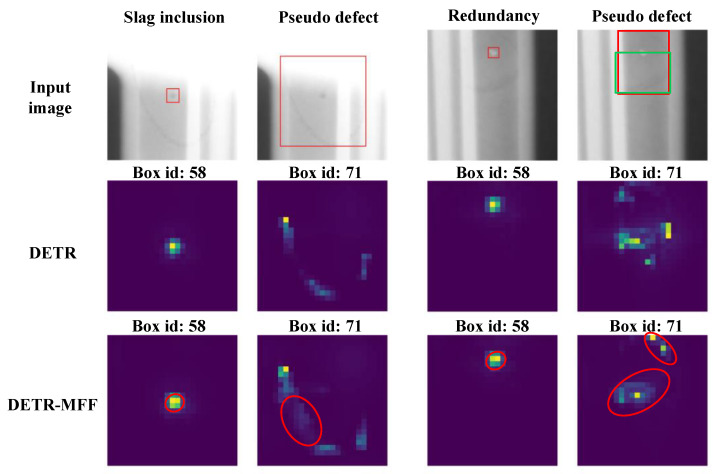
Sample visualization of the multi-scale defect analysis from the DETR and DETR-MFF model.

**Figure 13 sensors-24-01663-f013:**
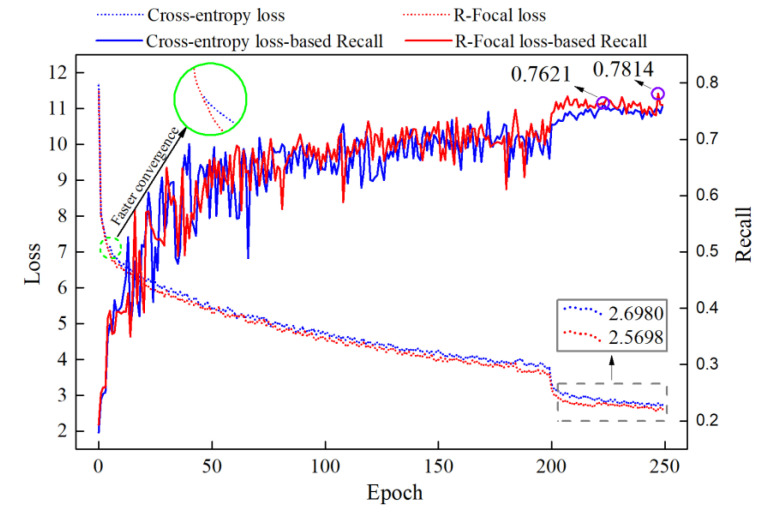
Loss curve and Recall curve with and without R-Focal loss.

**Figure 14 sensors-24-01663-f014:**
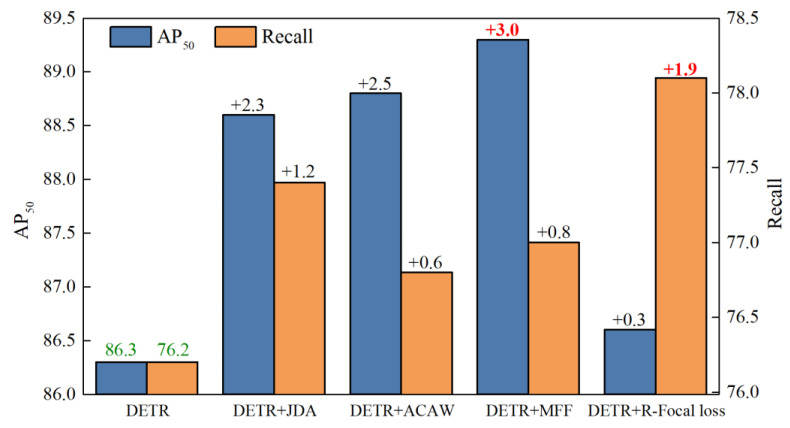
Improvement of AP and Recall varied between different modules.

**Table 1 sensors-24-01663-t001:** Statistical results of defect types.

Types of Defects	Slag Inclusion	Redundancy	Pseudo Defect	Cold Shut	Gas Cavity	Crack	Sum
Proportion (%)	62.7	21.1	7.2	5.3	2.0	1.7	100

**Table 2 sensors-24-01663-t002:** Hardware and software parameters of experiment.

Hardware	CPU	Intel (R) Xeon Gold 6226R @ 2.90 GHz
	RAM	Samsung DDR4 16 GB ×8
	GPU	NVIDIA GeForce RTX 3090 GPU
Software	OS	Windows 10
	CUDA Toolkit V11.7	
	CUDNN V8.4.0	
	Python 3.8	
	Torch 2.0.0; Torchvision 0.15.1	

**Table 3 sensors-24-01663-t003:** Varying *a* and *b* for JDA (*c* = 0).

a	b	AP	${AP}_{50}$	${AP}_{75}$
0	0	60.2	86.3	68.2
0	50	60.5	86.6	68.5
20	40	60.9	86.9	68.7
20	30	61.2	87.2	68.7
30	30	**62.0**	**87.8**	**69.8**
30	20	61.8	87.5	69.2
40	20	61.0	86.9	68.8
50	0	60.9	86.5	68.3

**Table 4 sensors-24-01663-t004:** Varying *c* for JDA (w. optimal *a* and *b*).

*c*	a	b	AP	${AP}_{50}$	${AP}_{75}$
1	30	20	61.9	87.8	69.3
2	30	30	62.3	88.1	70.1
5	30	30	**62.4**	**88.6**	**71.2**
7	30	30	62.1	88.2	**71.2**
9	40	20	60.8	87.2	68.8
10	40	20	60.4	86.9	68.5

**Table 5 sensors-24-01663-t005:** Parameters and mAP of different backbones with ACAW module.

Module	Backbone	Params (M)	GFLOPs	AP	${AP}_{50}$	${AP}_{75}$
Without ACAW	ResNet-50	41.28	6.93	60.2	86.3	68.2
	ResNet-101	60.22	11.78	61.6	88.6	70.5
	VGG-16	152.03	20.12	52.2	76.1	60.8
ACAW	ResNet-50	64.48	8.74	61.1	88.8	70.1
	ResNet-101	83.42	13.60	61.9	89.3	71.1
	VGG-16	175.23	24.21	**54.6**	**79.4**	**62.9**

**Table 6 sensors-24-01663-t006:** Comparison results of mAP between traditional methods and proposed DETR-MFF.

Method	AP	${AP}_{50}$	${AP}_{75}$	${AP}_{S}$	${AP}_{M}$	${AP}_{L}$
SSD	54.2	87.2	58.6	50.4	48.2	63.4
YOLOv5-L	60.9	87.8	66.2	55.7	48.7	85.4
DETR	60.2	86.3	68.2	52.4	47.7	88.3
Deformable DETR	62.8	88.1	70.8	**56.4**	48.6	78.2
DETR-MFF	**63.1**	**89.3**	**71.5**	55.8	**49.0**	**94.5**

**Table 7 sensors-24-01663-t007:** Comparison results of R-Focal loss under various hyper-parameters with different epochs.

γ	α	R-Focal Loss Value				
		Epoch = 10	Epoch = 15	Epoch = 20	Epoch = 25	Epoch = 30
0	0	6.74	6.50	6.28	6.07	6.03
0.2	0.5	7.26	7.14	6.90	6.72	6.34
1.0	0.5	7.15	7.04	7.01	6.67	6.24
1.0	0.25	6.89	6.62	6.51	6.10	6.01
2.0	0.25	6.72	6.48	6.21	6.01	5.82
5.0	0.25	**6.63**	**6.41**	**6.14**	**5.93**	**5.75**

**Table 8 sensors-24-01663-t008:** Comparison results of detection mAP between the proposed method and other traditional methods.

Methods	Backbone	${AP}_{S}$	${AP}_{M}$	${AP}_{L}$	${AP}_{GC}$	${AP}_{C}$
SSD	VGG-16	50.4	48.2	63.4	30.2	80.2
Faster-RCNN	ResNet-50	26.3	58.2	82.6	80.2	57.6
	VGG-16	40.2	56.2	62.5	78.0	56.7
YOLOv5-L	CSPDarknet-53	55.7	48.7	85.4	40.3	41.8
DETR	ResNet-50	52.4	47.7	88.3	51.2	60.8
Deformable DETR	ResNet-50	56.4	48.6	78.2	45.2	65.8
CenterNet	Hourglass-104	58.7	49.2	90.7	78.9	80.4
Proposed method	ResNet-50	**61.2**	**59.7**	**97.5**	**92.7**	**88.6**

**Table 9 sensors-24-01663-t009:** Defect detection performance of different methods on the ATBDX dataset.

Methods	Backbone	*F*1	*Recall*	AP	${AP}_{50}$
SSD	VGG-16	83.7	79.2	54.2	87.2
Faster-RCNN	ResNet-50	68.2	80.5	55.4	89.6
	VGG-16	72.1	76.1	59.4	90.3
YOLOv5-L	CSPDarknet-53	78.5	72.2	60.9	87.8
DETR	ResNet-50	79.3	73.1	60.2	86.3
Deformable DETR	ResNet-50	81.5	74.6	62.8	88.1
CenterNet	Hourglass-104	81.2	78.4	64.8	90.6
Proposed method	ResNet-50	**85.6**	**80.8**	**67.2**	**92.8**

## Data Availability

For confidentiality reasons, no further details will be disclosed at this time.

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
