# Peer review of "Detection Transformer with Multi-Scale Fusion Attention Mechanism for Aero-Engine Turbine Blade Cast Defect Detection Considering Comprehensive Features"

_sensors, 2024, doi:10.3390/s24051663_

Round 1
Reviewer 1 Report
Comments and Suggestions for Authors
Dear Authors,
Thank you for submitting your manuscript to our journal. Your dedication to advancing the field is appreciated.
While the manuscript shows promise, I have identified areas for improvement. Your work holds great potential, and I look forward to a revised version that will enhance its clarity and impact.
My comments:
1. what is the data size of images used before and after augmentation?
2. Small-size defects were mentioned; what is the smallest size that needs to be detected and is considered a crack or another type of defect?
3. Discuss an engine's life cycle and quantify time savings achieved through your method. Relate these details to practical applications for a more impactful presentation. How can the industry apply such a method?
4. Include more references from Sensors to enhance the paper's scholarly foundation. I have seen only one article in the list.
5. Provide a concise explanation of aero engine degradation in the introduction to improve reader understanding.
Comments on the Quality of English LanguageMinor Edits required
Reviewer 2 Report
Comments and Suggestions for Authors
The article proposes a novel method for detecting casting defects in aeroengine turbine blades using a detection transformer with multi-scale fusion attention mechanism. The proposed method addresses challenges such as multi-scale defect detection and imbalanced datasets by incorporating a joint data augmentation approach and an attention-based feature enhancement module. The multi-scale feature fusion module integrates high-dimensional semantic information and low-level representation features to enhance defect detection precision. The proposed method is validated using an aeroengine turbine blade defect X-ray image dataset and is shown to outperform traditional methods in terms of detection performance.However, there are some problems, such as references, the explanation of the principle and grammar, etc. Detailed comments are listed as follows. Hope they can provide help for improving the quality of this paper.
1) The references to this article are insufficient. The following references are recommended to be added.
(1)Lei Yang*, Jian Yu, Yanjie Guo, Sicheng Chen, Kunpeng Tan, Sheng Li, An electrode-grounded droplet-based electricity generator (EG-DEG) for liquid motion monitoring, Advanced Functional Materials, 2023, DOI: 10.1002/adfm.202302147
(2)What are the challenges faced by traditional methods in detecting casting defects in aeroengine turbine blades?
(3)How does the proposed joint data augmentation approach address the issue of imbalanced datasets in blade defect detection?
(4)What is the attention-based channel adaptive weighting (ACAW) feature enhancement module and how does it contribute to the defect detection process?
(5)How does the multi-scale feature fusion module integrate high-dimensional semantic information and low-level representation features to enhance defect detection precision?
(6)What evaluation metrics were used to validate the effectiveness of the proposed method, and how did it perform compared to traditional methods?
Reviewer 3 Report
Comments and Suggestions for Authors
This is the comments on the Manuscript to: Sensors (ISSN 1424-8220)
Manuscript ID: sensors-2848803
Type of manuscript: Article
Title: Detection transformer with multi-scale fusion attention mechanism for aeroengine turbine blade cast defect detection considering comprehensive features.
Author: Han-Bing Zhang , Chun-Yan Zhang , De-Jun Cheng, Kai-Li Zhou , Zhi-Ying Sun.
Rate the Manuscript:
Significance to field and specialization of “Sensors (ISSN 1424-8220)” journal: good.
In the article it has been presented the results of the investigation of casting defect detection which plays a crucial role in improving aircraft performance. Existing defect detection methods face the challenges in effectively detecting multi-scale defect and handling imbalanced datasets, which leads to unsatisfactory defect detection results.
The main conclusions:
In this work propose a novel blade defect de-tection method based on detection transformer with multi-scale fusion attention mechanism with considering comprehensive features.
1. Novel joint data augmentation (JDA) method is constructed to alleviate the imbalanced datasets issue, which effectively increases the number of sample data.
2. An attention-based channel adaptive weighting (ACAW) feature enhancement module is established to fully utilize complementary information among different feature channels, and further refine feature representations.
3. A multi-scale feature fusion (MFF) module is proposed to integrate high-dimensional semantic information and low-level repre-sentation features, thus enhancing multi-scale defect detection precision.
4. Scientific content: good.
Originality: good.
Clarity and presentation: acceptable.
Appropriateness for Journal: appropriate subject matter for the “Sensors (ISSN 1424- 8220)”
Need for rapid publication: no.
1. What is the main question addressed by the research? An aeroengine turbine blade defect X-ray (ATBDX) image dataset were applied to validate the proposed method.
2.Do you consider the topic original or relevant in the field? Does it
address a specific gap in the field?
Yes.What does it add to the subject area compared with other published material?
3.Yes. The comparative results demonstrate that the proposed method can effectively integrate multi-scale image features and improve multi-scale defect detection precision.
4.What specific improvements should the authors consider regarding the methodology? What further controls should be considered?
The research began by analyzing the R-Focal loss is developed in the MFF attention-based detection transformer (DETR) to further solve the issue of imbalance dataset and accelerate model convergence by random hyper parameters search strategy.
5. Are the conclusions consistent with the evidence and arguments presented and do they address the main question posed?
Yes.
6. Are the references appropriate?
· Yes. The references are appropriate.This research based on 45 scientific works. May be consider as additional references: Diagnostics of degradation of the lock of a sealing ring according to the loss of working gases of an internal combustion engine // Materials Science. - 2014. – Vol.50, N 1. – P. 156 -159. DOI 10.1007/s11003-014-9704-9
7. Additional comments on the tables and figures and remarks.
· Who conducted the DS-Cascade RCNN with 73 deformable convolution to detect hub defects, which can adaptively adjust the position 74 and size of the convolution kernel with considering the defect shape (lines 73-74) and in what way was it inadequate?
· Fig. 4 shows a large difference in the overall architecture of the proposed MFF attention-based DETR. What is the reason for this?
· The manuscript should be checked by native speaker for correct grammar and spelling.

This is the comments on the Manuscript to: Sensors (ISSN 1424-8220)
Manuscript ID: sensors-2848803
Type of manuscript: Article
Title: Detection transformer with multi-scale fusion attention mechanism for aeroengine turbine blade cast defect detection considering comprehensive features.
Author: Han-Bing Zhang , Chun-Yan Zhang , De-Jun Cheng, Kai-Li Zhou , Zhi-Ying Sun.
Rate the Manuscript:
Significance to field and specialization of “Sensors (ISSN 1424-8220)” journal: good.
In the article it has been presented the results of the investigation of casting defect detection which plays a crucial role in improving aircraft performance. Existing defect detection methods face the challenges in effectively detecting multi-scale defect and handling imbalanced datasets, which leads to unsatisfactory defect detection results.
The main conclusions:
In this work propose a novel blade defect de-tection method based on detection transformer with multi-scale fusion attention mechanism with considering comprehensive features.
1. Novel joint data augmentation (JDA) method is constructed to alleviate the imbalanced datasets issue, which effectively increases the number of sample data.
2. An attention-based channel adaptive weighting (ACAW) feature enhancement module is established to fully utilize complementary information among different feature channels, and further refine feature representations.
3. A multi-scale feature fusion (MFF) module is proposed to integrate high-dimensional semantic information and low-level repre-sentation features, thus enhancing multi-scale defect detection precision.
4. Scientific content: good.
Originality: good.
Clarity and presentation: acceptable.
Appropriateness for Journal: appropriate subject matter for the “Sensors (ISSN 1424- 8220)”
Need for rapid publication: no.
1. What is the main question addressed by the research? An aeroengine turbine blade defect X-ray (ATBDX) image dataset were applied to validate the proposed method.
2.Do you consider the topic original or relevant in the field? Does it
address a specific gap in the field?
Yes.What does it add to the subject area compared with other published material?
3.Yes. The comparative results demonstrate that the proposed method can effectively integrate multi-scale image features and improve multi-scale defect detection precision.
4.What specific improvements should the authors consider regarding the methodology? What further controls should be considered?
The research began by analyzing the R-Focal loss is developed in the MFF attention-based detection transformer (DETR) to further solve the issue of imbalance dataset and accelerate model convergence by random hyper parameters search strategy.
5. Are the conclusions consistent with the evidence and arguments presented and do they address the main question posed?
Yes.
6. Are the references appropriate?
· Yes. The references are appropriate.This research based on 45 scientific works. May be consider as additional references: Diagnostics of degradation of the lock of a sealing ring according to the loss of working gases of an internal combustion engine // Materials Science. - 2014. – Vol.50, N 1. – P. 156 -159. DOI 10.1007/s11003-014-9704-9
7. Additional comments on the tables and figures and remarks.
· Who conducted the DS-Cascade RCNN with 73 deformable convolution to detect hub defects, which can adaptively adjust the position 74 and size of the convolution kernel with considering the defect shape (lines 73-74) and in what way was it inadequate?
· Fig. 4 shows a large difference in the overall architecture of the proposed MFF attention-based DETR. What is the reason for this?
· The manuscript should be checked by native speaker for correct grammar and spelling.
Reviewer 4 Report
Comments and Suggestions for Authors
The authors have proposed the following manuscript: "Detection Transformer with Multi-Scale Fusion Attention Mechanism for Aero engine Turbine Blade Cast Defect Detection Considering Comprehensive Features "in which they develop a blade defect detection method based on detection transformer with multi-scale fusion attention mechanism which integrates 765 JDA module, ACAW feature enhancement module, MFF module and MFF attention- 766 based DETR
The manuscript is quite well written and structured, the effectiveness of the proposed method being demonstrated by comparing it with other methods in terms of different evaluation indicators. The comparison shows that the proposed method has better detection performance than the other traditional methods, which proves that it has the ability to take into account the diversity of imbalanced data set to develop a reliable defect detection model.
Round 2
Reviewer 2 Report
Comments and Suggestions for Authors
I have no questions